# Study on the Influence of Wet Backfilling in Open Pit on Slope Stability

**Qiusong Chen, Yufeng Niu and Chongchun Xiao ***

School of Resources and Satety Engineering, Central South University, Changsha 410083, China;
qiusong.chen@csu.edu.cn (Q.C.); 8210203208@csu.edu.cn (Y.N.)
* Correspondence: 22022135@csu.edu.cn

**Abstract:** The residual open pit left in the wake of open-pit mining poses significant safety hazards, with backfilling being an effective strategy to wholly eliminate these risks. The stability of the slope following wet backfilling, however, should not be overlooked. This paper examines the impact of the seepage field conditions and backfill height on the stability of open-pit slopes using a case study of cemented backfill in a specific open pit in Anhui Province. Moreover, it utilizes onsite research, Slide simulations, and similar simulation tests. The study findings suggest that as the height of the tailing solidification backfill increases, the safety factor of open-pit slopes gradually elevates. When the backfill height exceeds 10 m, all profiles of the studied open-air slope fulfill the stability prerequisites. Furthermore, when the solidification backfill reaches 20 m, all profiles of the studied open-pit slope satisfy the stability requirements. The research outcomes offer a methodology for mining corporations to avert slope instability and destruction, thereby providing effective safeguards for the extraction of scarce resources in mines.

**Keywords:** wet backfilling; slope stability; open-pit mining

## 1. Introduction

Open-pit mining has contributed significantly to China's economy. However, several open-pit mines have been forced to close due to their failure to adhere to sustainable mining practices for mineral resources [1], resulting in substantial open pits and steep slopes that pose significant safety risks [2]. The principle of "whoever damages, recovers" has been proposed in numerous countries, including China, leading to the restoration and management of open-pit mines becoming a subject of great interest. The reclamation and comprehensive utilization of open-pit mines generally adopt the following approaches: (1) backfilling, (2) slope trimming and reinforcement, and (3) comprehensive utilization [2–4]. The backfilling technique not only resolves the issue of tailing disposal but also mitigates safety risks in the open pit. This approach economizes the costs of open-pit reclamation and balances the economic benefits and engineering safety. The prevalent method of backfilling open pits is performed through a process known as wet backfilling [5,6]. This process involves transporting the solid waste, such as tailings from the selection plant, around the open pit to a mixing system, combining it with water from the cementitious material warehouse to form a concentrated slurry [7,8]. This slurry is transported to the open pit for solidification and backfilling by natural flow or pumping. However, wet backfilling invariably filters out some water, prompting the question of whether local bleeding and soaking might compromise the stability of the open-pit slope. This uncertainty limits the application of the technology. Moreover, the slope failure has been recognized as the main safety risk in a closed open-pit mine. Thus, it is necessary to study the influence of wet backfilling in open pits on slope stability. Academic researchers have made significant progress in the field of slope stability using different methodologies, including deterministic analysis methods [9], numerical analysis methods [10], random analysis methods [11], and fuzzy analysis methods [12]. This article employs deterministic analysis methods for research

purposes. The limit equilibrium method is a quantitative and deterministic analysis method with the advantages of straightforward principles, a simple calculation method, and ease of understanding. It is the most well-established and oldest method among all quantitative analysis methods [13]. This research uses the version 6.0 of the Slide software, a tool developed by the Canadian company RocScience based on limit equilibrium theory, to analyze the stability of rock and soil slopes. This software enables the quick creation and analysis of complex models, performing slope stability analysis and probability analysis under various external loads or supports. In addition, this article further analyzes the theoretical and mathematical models associated with seepage, conducting simulation analyses of slope stability under the influence of a seepage flow field [14,15]. Currently, few studies combine simulation calculations with similar physical model experiments to investigate changes in slope stability during the open-pit backfilling process. Accordingly, this study focuses on a slope in Anhui, China, as its research subject. Upon completion of open-pit mining at a level of $-156$ m to form the final slope, the overall slope angle of the hanging wall slope is approximately $45°$. The surface elevation is approximately $+36$ m. The top of the footwall slope is the peak of Jitou Mountain. As the footwall slope extends to the base of the $-156$ m pit, the overall slope angle of the final slope ranges between $39°$ and $41°$, with the final vertical height of the footwall slope reaching 482 m, an unusually high and steep slope. The rock mass has been exposed for a long time and weathered severely. Under the erosion of slope rainfall and the damage and destruction of the rock mass caused by production blasting, multiple slope steps have collapsed and failed to preserve the platform, especially the high and steep slope of the footwall. When it softens with water and the mechanical strength decreases, landslides are prone to occur, causing the designed safety and cleaning platform to slide and collapse, forming a large range of smooth slopes, and the overall stability of the slope is affected. This study aims to mitigate the safety hazards in open-pit excavation by employing tailing-cemented backfill. Furthermore, it analyzes the impact of the seepage field and backfill height on the stability of open-pit slopes through field research and Slide simulation. Within the study context, a physical model is constructed based on similarity theory to monitor and verify the slope stability.

## 2. Materials and Methods

### 2.1. Slope Rock Mechanics Test

The sampling location was the footwall slope of the open pit, with the rock type being quartz sandstone. After processing the original ore and rock samples retrieved from the site into standard samples in the laboratory, the rock samples underwent saturation treatment. The soaking water served as the upper overflow water following the flocculation and settling of the tailings. The post-measurement, which concerned the average block density of the rock sample post-saturation treatment, was 2.94 tons/m$^3$. The uniaxial compressive strength, elastic modulus, Poisson's ratio, tensile strength, cohesion, and internal friction angle were measured using the SANS SHT4206 electro-hydraulic servo universal testing machine and UT7116 static strain gauge, among other equipment. The relevant physical properties of the post-saturation treatment are provided in Table 1.

**Table 1.** Related physical properties of rock samples after saturation treatment.

| Properties | Value |
| :---: | :---: |
| Uniaxial compressive strength | 112.91 MPa |
| Elastic modulus | 34.52 GPa |
| Poisson's ratio | 0.265 |
| Tensile strength | 12.67 MPa |
| Cohesive force | 0.53 MPa |
| Interior friction angle | 41.89° |

An image of the samples is provided below as Figure 1. We processed the ore samples collected from the field into standard specimens in the laboratory. To ensure the accuracy and reliability of the experimental results, we conducted tests using four standard specimens for each mechanical parameter measurement. The obtained numerical values for the mechanical parameters represented the average values of the respective specimens. It is worth noting that the mechanical parameters of the specimens showed minimal differences among them.

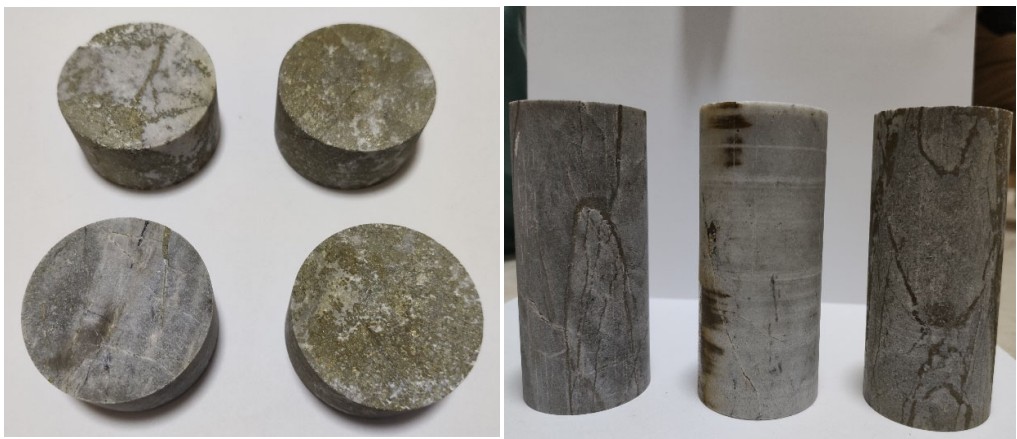

**Figure 1.** Images of test samples.

We compared the saturated mechanical parameters of quartz sandstone under different water quality levels, and the results are shown in the Table 2. From the table, it can be observed that, compared to the condition of water infiltration, the mechanical parameters affecting slope stability, such as the cohesive strength and internal friction angle, slightly decreased under the condition of tailwater immersion.

**Table 2.** Comparison of saturated mechanical parameters of quartz sandstone under different water quality levels.

| Parametric Test | Density (t/m$^3$) | Cohesive Force (MPa) | Angle of Internal Friction (°) | Elastic Modulus (MPa) | Poisson's Ratio |
|---|---|---|---|---|---|
| Soaking in water | 2.70 | 0.576 | 37.51 | 38.00 | 0.260 |
| Tailwater soaking | 2.94 | 0.530 | 36.39 | 36.77 | 0.265 |

### 2.2. Slope Stability Analysis Conditions

In reference to the "Seismic Parameter Zoning Map of China" (GB 18306-2015) [16], the seismic fortification intensity in the mining area was established as seven degrees, with the foundational seismic acceleration value set to 0.1 g. The seismic inertia force influence coefficient for each block was calculated when assessing seismic stability as outlined in Appendix D.2.1 of the Technical Specification for Slope Engineering of Non-Coal Open Pit Mines (GB 51016-2014) [17].

$$K_c = \alpha \xi \beta_i \tag{1}$$

where $\alpha$ is the seismic design acceleration based on the local horizontal seismic coefficient, which is the ratio of the statistical average for the maximum horizontal ground acceleration to the gravitational acceleration; $\xi$ is the reduction coefficient, taken herein as 0.25; $\beta_i$ is the dynamic distribution coefficient of the ith block, which is taken as 1.5 for this stability analysis.

The ultimate designation of the horizontal seismic inertia force coefficient for mine design is $Kc = 0.0375$, with the vertical seismic inertia force coefficient generally assumed to be 65% of the horizontal seismic inertia force coefficient. Hence, the research value was 0.024. The equivalent static load vibration coefficient was set to 0.025, considering the impact of the maximum horizontal dynamic load acting on the slope towards the mining

site. The slope safety factor represents the final quantitative measure in assessing slope stability. The safety grade of the open-pit mine slope engineering was decided based on the extent of slope damage and slope height. This factor was categorized into three levels: I, II, and III (refer to attached Tables S1 and S2 for details). Current regulations specify the safety factor of the slope, as depicted in Table 3. In this table, Condition I considers only the influence of self-weight stress, Condition II considers only the impact of self-weight and blasting vibration force, and Condition III considers the impact of self-weight and seismic force. Given the specific circumstances of the slope and the principle of prioritizing safety, the permissible safety coefficient of the slope was determined in this study, as shown in Table 4.

**Table 3.** Design safety factor of the overall slope under different working conditions.

| Safety Level of Slope Engineering | Safety Factor for Slope Engineering Design | | |
|---|---|---|---|
| | Condition I | Condition II | Condition III |
| I | 1.25–1.20 | 1.23–1.18 | 1.20–1.15 |
| II | 1.20–1.15 | 1.18–1.13 | 1.15–1.10 |
| III | 1.15–1.10 | 1.13–1.08 | 1.10–1.05 |

**Table 4.** Safety factor of the overall slope.

| Safety Level of Slope Engineering | Condition I | Condition II | Condition III |
|---|---|---|---|
| I | 1.25 | 1.23 | 1.20 |

*2.3. Analysis for Open-Pit Slope*

In order to facilitate monitoring and management, the previous mining operation divided the upper and lower slope profiles into sections, depicted in Figure 2. In this study, fourteen profile lines (A–H profiles of the upper wall and 1–6 profiles of the lower wall) were generally arranged vertically within a closed circular range from the eastern end of the upper wall slope to the eastern end of the lower wall slope. Preliminary site investigation and monitoring results indicated subpar stability for the upper wall slope C, E, and lower wall slope. Consequently, the analysis profiles selected were upper wall slope C, E profiles and lower wall slope sections 1–6. The backfilling plan for the open pit comprised the following: the pit's bottom was backfilled at 10 m (with the 28-day uniaxial compressive strength of the solidified body 5 m above the pit bottom not less than 3.0 MPa, and the strength of the solidified body at 5–10 m not less than 2.0 MPa). Above the pit bottom, regular backfilling was employed at 10 m (the strength of the solidified body was not less than 0.5 MPa). The feasibility of dry pile consolidation could be considered later based on the backfilling effectiveness and underground ground pressure monitoring data. After backfilling to a depth of 110–150 m above the pit bottom, a further increase in backfilling height will have a negligible impact on the safety of underground mining. Consequently, for this open-pit slope stability analysis, the solidification and backfilling heights were 0, 10 (bottom), 80, and 140 m. The backfill layout is displayed in Figure 3.

*2.4. Stability Simulation Calculation Method*

The safety of exposed slopes at varying backfill heights was examined using the Slide software version 6.0. The simulation process employed the JanBu method, which is an optimized non-arc approach [18]. The JanBu method posits the presence of horizontal and vertical forces between each rock layer. The computation process was significantly simplified due to the non-utilization of torque balance equations to solve. This method counters the disadvantage of assuming an arc-shaped failure surface as the condition, with no restrictions on the shape of the failure surface. The utilization of the Slide software for slope stability analysis involves two steps: one is to determine the anti-slip stability safety factor on a specific sliding surface within the landslide body, and the second is

to identify the critical sliding surface with the corresponding minimum safety factor among all potential sliding surfaces. The backfilling concentration of tailings paste is high, significantly reducing the bleeding rate of the solidified body to less than 5%. A minimal amount of bleeding may still occur, accumulating on the surface of the solidified body, while pipe-washing water before and after backfilling also enters the open pit. A slight amount of water accumulating on the surface of the solidified body benefits the development of the strength of the solidified body and the suppression of dust. However, water accumulated in the open pit can permeate the slope through cracks, impairing the performance of adjacent rock masses and thus affecting the overall slope stability. In addition, despite the effectiveness of the anti-seepage curtain grouting implemented on the periphery of the hanging wall slope in impeding groundwater, it cannot fully obstruct the hydraulic connection inside and outside the curtain. The influence of groundwater on slope stability, particularly the hanging wall slope, cannot be disregarded due to the contribution of backfill filtration water. This study sought to calculate a series of safety factors for slip surfaces through the Slide software and employ different methods to locate critical slip surfaces that satisfied the requirements. When conducting a statistical probability analysis, parameters for statistical distribution, such as the material properties, support characteristics, and loads, are defined. Their uncertainty is interpreted by assigning statistical distribution types to one or more parameters in the model, and the probability (or reliability index) of a slope landslide is computed. The probability analysis of slope stability supplements and enhances the traditional deterministic (safety factor) analysis method, and substantial valuable conclusions can be drawn from the probability analysis of slopes. The same model as in the slope stability analysis was utilized to define and analyze groundwater problems through the "Groundwater Analysis Roundwater Analysis" in Slide. The defined boundaries were usable for both groundwater analysis and slope stability analysis. In other words, upon completion of the groundwater analysis, the computation results were automatically applied for slope stability analysis.

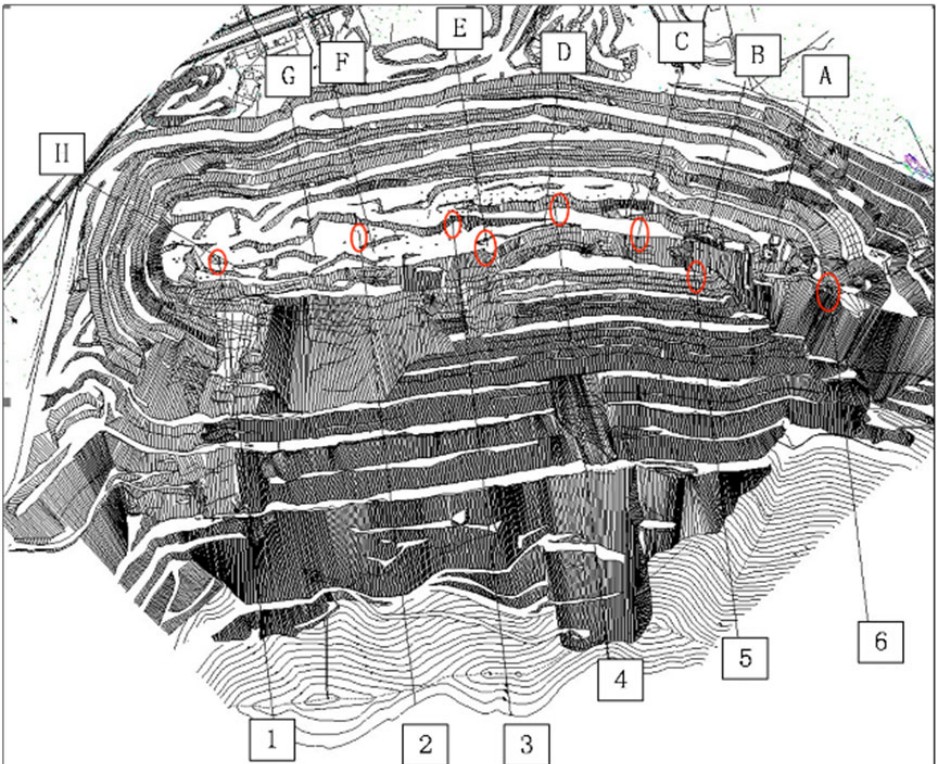

**Figure 2.** Section division of the upper and lower walls of the slope.

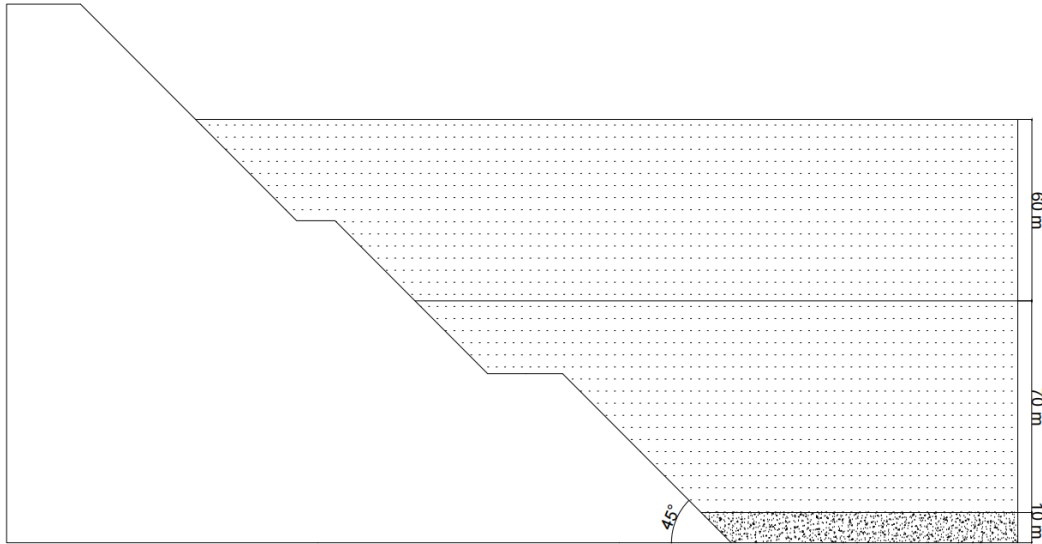

**Figure 3.** Layout plan for backfilling of open pit.

*2.5. Similarity Model Test*

The process of slope stability analysis has somewhat simplified the boundary conditions. Consequently, this study employed a similar physical model to conduct strain testing on the waterlogged open-pit slope in order to further validate the reliability of the numerical simulation results. The primary objective of the model test was to replicate the impact of solidification and backfilling in the open pit on the safety performance of the upper and lower wall slopes. Therefore, the middle section of the open pit was selected for a similar simulation based on the actual occurrence form of the ore body and its relative position with the open pit. Due to the significant height of the footwall slope, the geometric similarity ratio (Cl) was determined to be 190, and a corresponding weight of an iron block was selected for similarity replacement. Materials of similarity were combined and mixed according to the ratio parameter to form a uniform slurry for layered pouring, with each layer having a height of 0.2 m. After pouring each layer, it was compacted and flattened to avoid bubbles in the model. Approximately 0.01 m of water was added to the bottom of the similar model pit every 0.5 h to crudely simulate the impact of backfill water on slope stability. The model test mainly monitored the strain situation of the upper and lower wall slopes during the solidification and backfilling process. The layout of the measurement points and the physical model are shown in Figure 4. The strain measurement points 1# and 2# were arranged on the upper slope of the open-pit model, while the strain measurement points 3# and 4# were arranged on the lower slope of the open pit, as shown in Supplementary Materials Table S3.

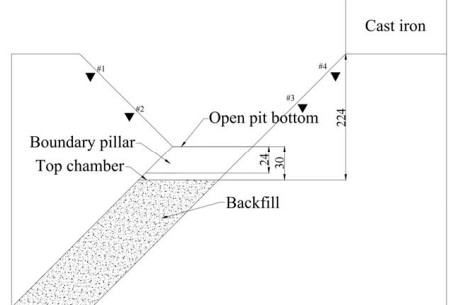
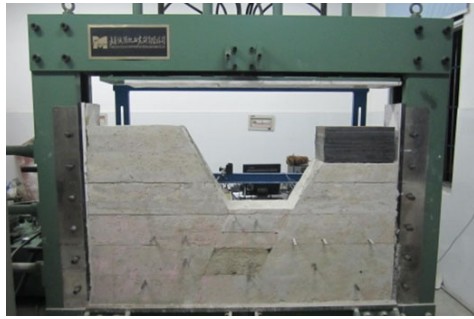

**Figure 4.** Design diagram of the simulation experimental measuring points and physical diagram of the simulation layout (unit: millimeters).

## 3. Results and Discussion

### 3.1. The Influence of Different Backfill Heights on Slope Stability under the Condition of No Seepage Flow Field

The calculation results of the safety coefficients of the open-air slopes C, E, 1, 2, and 3 of the hanging wall slopes are displayed in Figures 5 and 6 under three non-seepage flow field conditions (Condition I considers only the self-weight, Condition II considers the self-weight plus blasting vibration force, and Condition III considers the self-weight plus seismic force). According to the Technical Specification for Slope Engineering of Coal-Free Open-Pit Mines (GB 51016-2014), the safety level of this slope is Level 1. The ultimate safety factor of the slope under Conditions I, II, and III is calculated using the principle of safety first as 1.25, 1.23, and 1.20, respectively. The following diagrams illustrate the slope safety coefficients of the upper wall profiles C, E and lower wall profiles 1–6 under different operating conditions, as well as the partial slide analysis diagrams under non-seepage flow conditions.

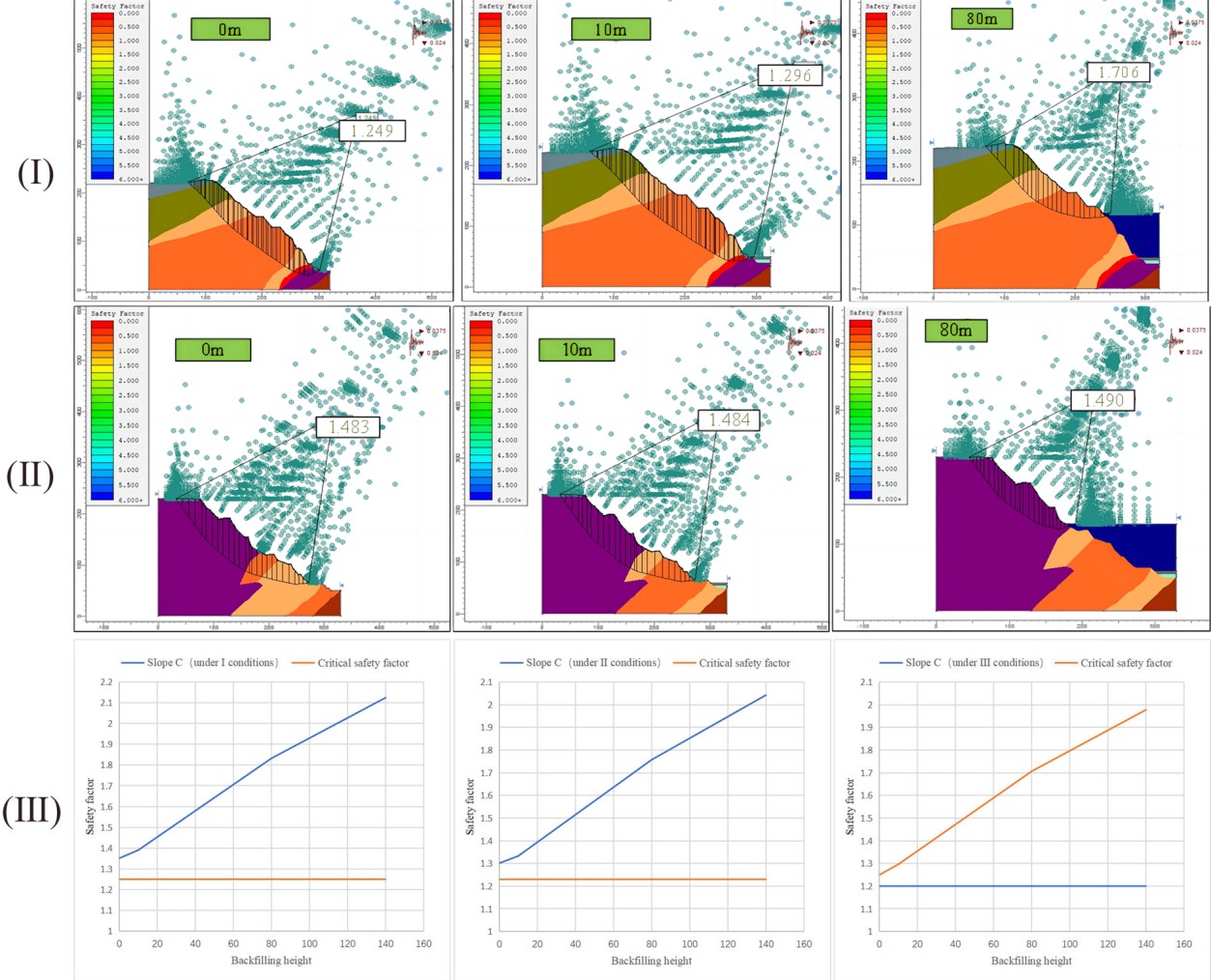

**Figure 5.** (**I**) Safety factor of C profile slopes under seismic Condition (**III**). (**II**) Safety factor of E profile slopes under seismic Condition (**III**). (**III**) Safety factor of slope C under normal working conditions (working Condition (**I**)), blasting working conditions (working Condition (**II**)), and seismic working conditions (working Condition (**III**)).

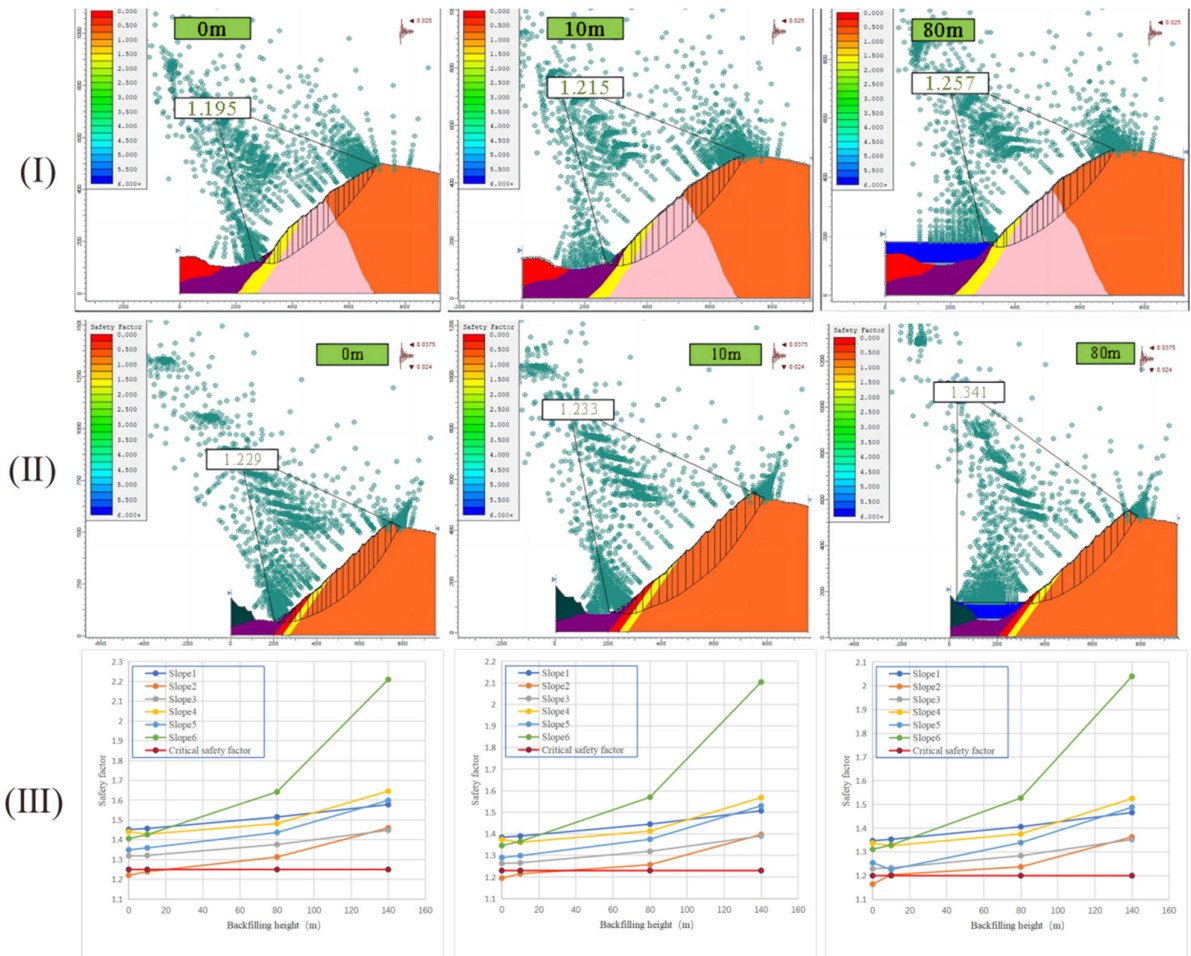

**Figure 6.** (**I**) Safety factor of section 2 slope under blasting Condition (**II**); (**II**) considering seismic condition (Condition (**III**)) 3 profile slope safety factor; (**III**) safety factor of the slope for sections 1–6 under normal working conditions (working Condition (**I**)), blasting working conditions (working Condition (**II**)), and seismic working conditions (working Condition (**III**)).

In summary, without considering the effect of the seepage field, the stability evaluation conclusions for open-pit slopes with different backfill heights are as follows.

(1) The safety factors of each section of the hanging wall slope meet the requirements of specifications and standards under the condition of non-solidified backfill. The stability of all sections in the footwall slope satisfies the requirements, and there are no safety hazards to the slope, except for the minimum safety factor of section 2, which is slightly below the critical value and poses potential landslide and collapse risks under long-term wind and rain erosion.

(2) As the height of tailing solidification and backfilling increases, the safety factor of open-pit slopes progressively increases. When the backfilling height exceeds 10 m, even the most unstable footwall slope profile 2 meets the stability requirements, eliminating local dangerous slope safety hazards. Therefore, carrying out open-pit solidification and backfilling is beneficial in improving slope stability.

(3) In the early stage of solidification and backfilling in the open pit (with a backfilling height of less than 10 m), appropriate reinforcement or disposal measures should be taken for the unstable parts, such as the slope near the profile of panel 2, to ensure the safety of the solidification and backfilling of tailings in the open pit.

### 3.2. Impact of Different Backfill Heights on Slope Stability under Seepage Field Conditions

Figure 7 shows a slide analysis of some sections under different operating cases considering seepage field conditions. As the backfill height increases and the load on the slope increases, the water channel between the slope and the pit bottom/surface decreases, thereby reducing the impact of water seepage on the stability of the slope [19–21]. In summary, considering the effect of the seepage flow field, the stability evaluation conclusions for open-pit slopes with different backfill heights are as follows.

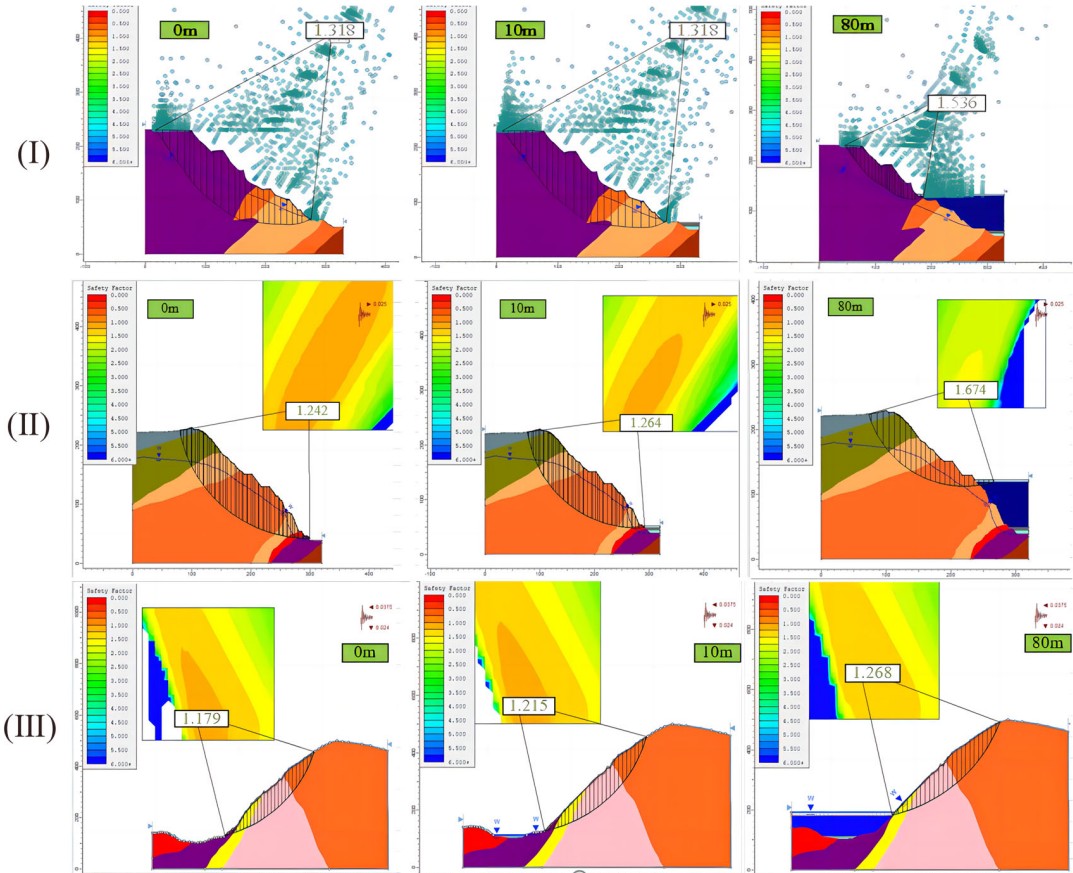

**Figure 7.** (**I**) Safety coefficient of the slope in section E under the action of seepage field (Condition (**I**)); (**II**) safety coefficient of the slope in section C under the action of seepage field (Condition (**II**)); (**III**) safety coefficient of the slope in profile 2 under the action of seepage field (Condition (**III**)).

(1)  When backfilling within 10 m, except for the upper wall slope profile C and the lower wall slope profile 2, where the minimum safety factor is slightly less than the critical value, the stability of all profiles meets the requirements, and there are no safety hazards to the slope.

(2)  As the height of tailing solidification and backfilling increases, the safety factor of open-pit slopes progressively increases. When the solidification and backfilling height reaches 20 m, all slope profiles considering a seepage flow field can meet the stability requirements, eliminating local dangerous slope safety hazards when not backfilling.

(3)  In the early stage of solidification and backfilling in the open pit (with a backfilling height of less than 20 m), appropriate reinforcement or disposal measures should be taken for areas with poor stability, such as the slopes near the C section of the upper wall slope and the 2 section of the lower wall slope, to ensure the safety of the solidification and backfilling of open-pit tailings.

*3.3. Analysis of Similar Physical Model Test Results for Slope Displacement Characteristics*

The results of the hanging wall slope analysis are shown in Figure 8.

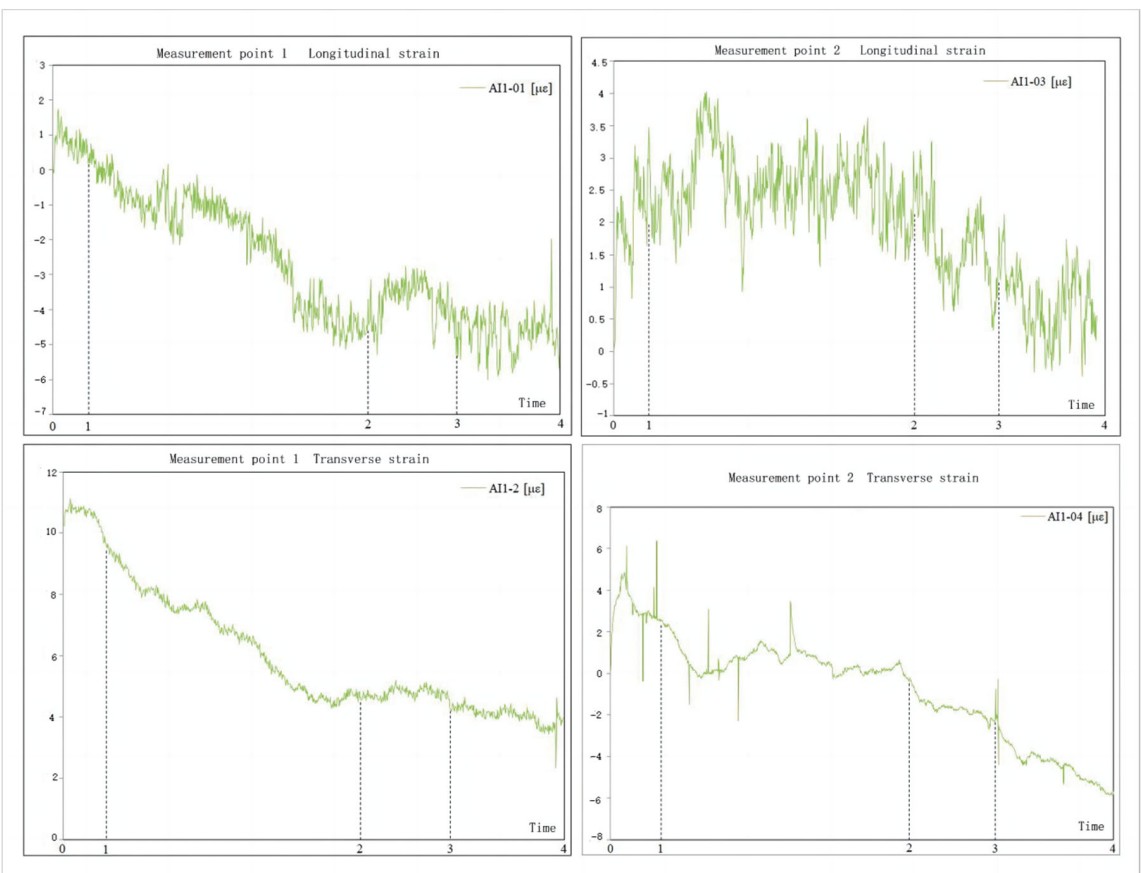

**Figure 8.** Longitudinal and transverse strains at measuring points 1 and 2.

(1)    The longitudinal strain at the upper slope of the hanging wall (measurement point 1) continually increases with the height of the backfill and gradually stabilizes in the later stage of backfilling (between stages 2 and 3, 110–150 m).

(2)    The longitudinal strain at the mid-point of the hanging wall slope (measurement point 2) demonstrates a minor fluctuation with the rise in backfill height during the 0–2 stage (0–10 m). Upon reaching the 2–3 stage (110–150 m), the strain value declines gradually, tending towards zero. This occurrence is possibly due to the strain gauge being obscured by the solidified matter in the later stages of backfilling.

(3)    The lateral strain at the upper slope of the hanging wall (measurement point 1) is more sensitive to changes in backfill height than the longitudinal strain, causing greater fluctuations in the strain value. The strain value decreases progressively with the advancement of the backfill operation. After reaching the mid-point of stages 1–2 (10–110 m), the strain value gradually stabilizes.

(4)    The transverse strain at the middle of the hanging wall slope (measurement point 2) diminishes with the increasing backfill height during the initial backfill stage. Upon entering the middle of stages 1–2 (10–110 m), the strain value rises inversely. This phenomenon might be attributed to the strain gauge being obscured by the solidified matter in the middle and later stages of backfilling.

The footwall slope analysis results are shown in Figure 9.

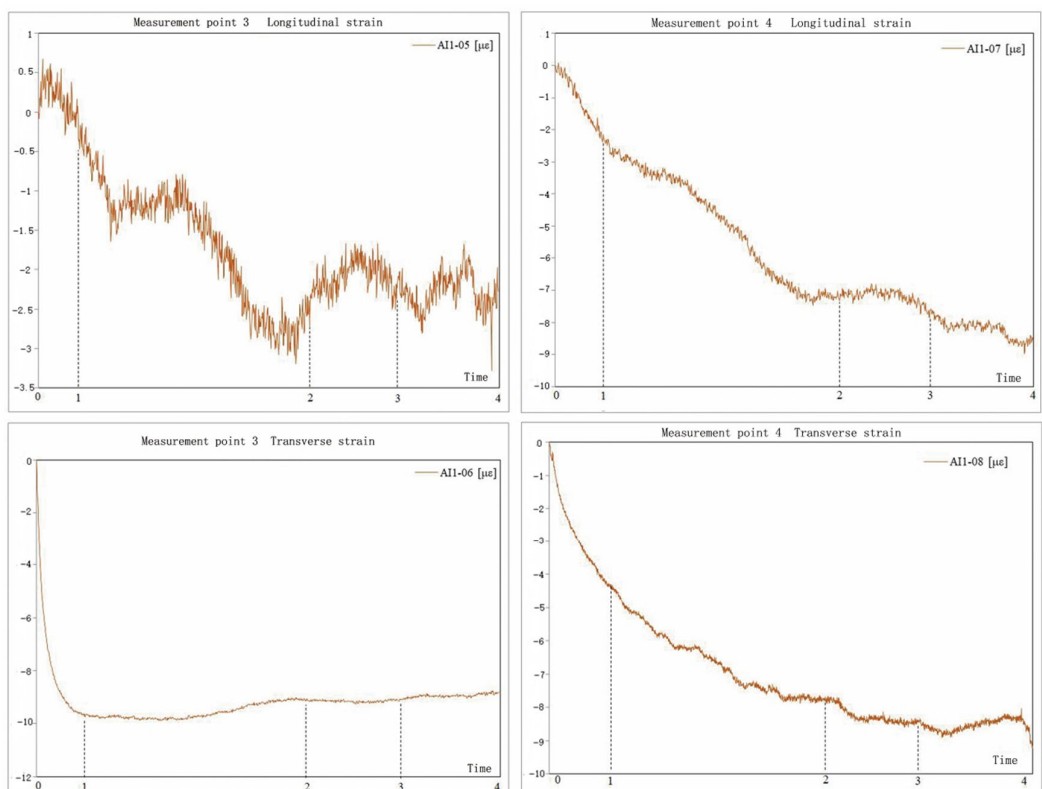

**Figure 9.** Longitudinal and transverse strains at measuring points 3 and 4.

(1) The longitudinal strain at the upper slope of the footwall (measurement point 3) increases slowly with the rising backfill height during the initial stage. After entering the middle and later zones of stages 1–2 (10–110 m), the strain value decreases slightly and maintains a relatively stable state.

(2) The longitudinal strain at the middle slope of the footwall (measurement point 4) rapidly increases with the rise in backfill height during the 0–2 stages (0–110 m). Upon entering stages 2–3 (110–150 m), the strain value fluctuates minimally while remaining relatively stable overall.

(3) The lateral strain at the upper slope of the footwall (measurement point 3) significantly increases with the rise in backfill height during the 0–1 stages (0–10 m). After entering stages 1–2 (10–110 m), the strain value remains relatively stable and slightly decreases overall.

(4) The lateral strain at the middle slope of the footwall (measurement point 4) is directly proportional to the backfill height and increases linearly with the backfill operation's progression. After entering stages 2–3 (110–150 m), the strain values start to stabilize.

## 4. Conclusions

This study aimed to investigate the stability of the open-air slope under varying solidification backfill heights and waterlogging conditions. Within this context, the mechanical parameters of the slope rock mass under soaking conditions were measured, and the stability of an open-air slope was analyzed by applying theoretical analysis, numerical simulation, and physical modeling. The main findings are as follows.

(1) After onsite sampling, sample processing, and indoor rock mechanics testing, the physical and mechanical indices of the sample, such as the block density, uniaxial compressive strength, elastic modulus, Poisson's ratio, and tensile strength, were measured under saturated conditions of tailing filtration. The results indicated that the mechanical parameters such as cohesion and the internal friction angle, which affect

slope stability, slightly decreased when the saturated rock samples were immersed in tailwater as compared to soaking in open-pit water.

(2) The limit equilibrium theory was applied using the Slide software to analyze the stability of exposed slopes under different solidification backfill heights without considering the effect of the seepage field. The research indicated that the safety factor of the open-pit slope increased as the tailing backfill height solidified. When the backfill height exceeded 10 m, all profiles of the studied open-pit slope met the stability requirements.

(3) The stability of exposed slopes under seepage was analyzed using the seepage field theory and Slide software. The research showed that as the height of tailing solidification backfill increased, the safety factor of the open-pit slope gradually increased. When the solidification backfill reached 20 m, all profiles of the studied open-pit slope met the stability requirements, thereby eliminating the potential safety hazards that might have existed in the non-solidification backfill.

(4) The strain at the upper and lower slopes of the open-pit at different solidification backfill heights was tested under waterlogged conditions using the physical similarity model of the open-pit slope backfilling operation. The research demonstrated that the displacement development law of open-pit slopes is consistent whether there is ponding or no ponding in open-pit pits. Hence, a small amount of ponding in open-pit pits has a minimal impact on the overall displacement of open-pit slopes.

(5) The studied mine was located in Anhui Province, China, where the average temperature was approximately 17 °C; thus, we did not consider the effects of high and low temperatures. However, the temperature is an important factor affecting the consolidation of backfilling, and we will further study this issue in the future.

In conclusion, reliable anti-seepage treatment should be administered based on the engineering survey results of the open pit before commencing solidification and backfilling operations. Other reinforcement measures should be applied depending on the slope's stability to ensure the open-pit slope's stability in the early stages of solidification and backfilling (before the backfilling height reaches 20 m). When the height of the solidified backfill exceeds 20 m, even considering the effect of the seepage field, the open-air slope is more stable than in the non-solidified backfill case, thereby eliminating the safety hazards of local slopes during non-solidified backfilling. Even after the solidification and backfilling operations, it is necessary to retain the existing open-pit slope displacement and deformation monitoring system and continue monitoring it periodically. This is to promptly detect any potential hazards and take appropriate safety measures in a timely manner, ensuring the stability of the open-pit slope. In future research, in addition to controlling the backfill water content and backfill height, efforts should also be made to enhance the strength and stability of the filling material, as well as the temperature effect. The development of harmless backfilling techniques should be pursued to further improve the quality of open-pit backfilling. Therefore, the solidification and backfilling of open-pit pits generally improve slope stability, reduce or even eliminate the long-term geological hazard safety risks of large open-pit slopes, and do not affect the safety of open-pit solidification and backfilling operations.

**Supplementary Materials:** The following supporting information can be downloaded at: https://www.mdpi.com/article/10.3390/su151612492/s1, Table S1: Slope Hazard Levels. Table S2: Classification of safety levels for slope engineering. Table S3: Measurement points and layout positions.

**Author Contributions:** Conceptualization, Q.C. and Y.N.; methodology, Q.C.; software, Q.C.; validation, Q.C., Y.N. and C.X.; formal analysis, Q.C.; investigation, Q.C.; resources, Q.C.; data curation, Q.C.; writing—original draft preparation, Y.N.; writing—review and editing, Q.C.; visualization, Y.N.; supervision, Y.N.; project administration, Q.C.; funding acquisition, Q.C. All authors have read and agreed to the published version of the manuscript.

**Funding:** This study was financially supported by the National Natural Science Foundation of Hunan Province, China (2022JJ30714).

**Institutional Review Board Statement:** Not applicable.

**Informed Consent Statement:** Not applicable.

**Data Availability Statement:** Not applicable.

**Conflicts of Interest:** The authors declare no conflict of interest.

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
