# Peer review of "Study on the Influence of Wet Backfilling in Open Pit on Slope Stability"

_sustainability, doi:10.3390/su151612492_

Round 1

Reviewer 1 Report

This article discusses the effect of seepage field conditions and backfill height on pit slope stability using the example of a cemented backfill in a specific pit in Anhui Province. The article has elements of new results and scientific novelty.

However, having studied the material of the study in more detail, I had several serious remarks that I would recommend the authors to eliminate:

1. In the introduction, I did not see the formulation of the problem, as well as the allocation of a previously unexplored area. What up to the authors is not sufficiently studied? What are the unresolved scientific questions?

2. Also, in the introduction, explain the following points more clearly: backfill drains water, which can affect slope stability. And what's next? The quarry is closed. What are the threats of water release? Why do you need to research it? What does it affect? Further in the sections of the article, you note the impact on underground mining. But I recommend doing this in the introduction, to add significance. This way the problem statement will be clearer.

3. I would like to receive an answer from the authors of the discussion question. How is paste backfilling carried out in a quarry under the influence of climatic conditions - rains, high temperatures in summer, winter? How are favorable conditions for the formation of backfill mass achieved?

In my opinion Figure 1 looks unrepresentative. Change the quality and make the drawing in color. There are a sufficient number of programs where you can imagine career voids. It is necessary to give explanations in the caption to Figure 1-6 and A-H.

5. Unfortunately, I did not see the problem statement for numerical simulation in the Slide program. How are the geometric dimensions of the model justified? What are the accepted boundary conditions? What is the Criterion? What physical and mechanical properties were introduced into the model. It is necessary to substantiate this in a separate subsection.

6. I did not see the justification and description of equivalent materials used in physical modeling.

7. How was backfill strength determined and how was it used in the model?

8. Unfortunately, the quality of figures 4, 5 and 6 leaves much to be desired. Absolutely not visible axes and measurement scale.

9. The results of subsection 3.3 are not convincing enough. There are no patterns, graphs and drawings with the results of physical modeling.

10. There is no binding of research to grants, scientific topics. This would show the relevance and significance of the study.

11. The references is poorly designed and insufficient (14 sources is not enough). I recommend doing a deeper scientific review and adding more modern scientific papers on the problem under consideration.

In the presented form, the article cannot be published. Serious changes and additions are needed. In my opinion, after serious corrections, the article can be recommended for publication.

Dear authors, do work on improving the article.

Reviewer 2 Report

some minor spelling and grammatical errors. Can be fixed when preparing the revision.

Reviewer 3 Report

The manuscript studies the impact of seepage field conditions and backfill height on the stability of open-pit slopes, using onsite research, Slide simulations, and similar simulation tests. The results are interesting and can be published in Sustainability after following questions or comments being addressed.

1. In the introduction. Please add some references about simulation analysis.

2. Please double-check the format problems. Such as, In Table 2, variable symbols should be marked in italics.

3. Please improve image quality, especially Fig.5 and 6.

4. The authors are suggested to analyze the influence of different backfill heights on slope stability under the condition of seepage field more in-depth.

5. It is recommended to describe the limitations and add emphasis on the future development in the conclusion

6. English language is good, but still needs to be adjusted to improve the readability.

Minor editing of English language required

Round 2

Reviewer 1 Report

I carefully read the updated version of the article.

The authors answered the discussion questions and adopted recommendations for the article.

I am quite satisfied.

I wish the authors success in future research.

Sincerely, Reviewer

Reviewer 2 Report

The authors addressed my concerns in the revised version.